# Predictors of Non-Sentinel Lymph Node Metastasis in Patients with Positive Sentinel Lymph Node in Early-Stage Cervical Cancer: A SENTICOL GROUP Study

**DOI:** 10.3390/cancers15194737

**Published:** 2023-09-26

**Authors:** Basile Pache, Matteo Tantari, Benedetta Guani, Patrice Mathevet, Laurent Magaud, Fabrice Lecuru, Vincent Balaya

**Affiliations:** 1Department Women-Mother-Child, Gynecology and Obstetrics Unit, Lausanne University Hospital (CHUV), 1005 Lausanne, Switzerland; 2University of Lausanne (UNIL), 1015 Lausanne, Switzerland; 3Gynecology Department, Fribourg University Hospital, University of Fribourg, 1700 Fribourg, Switzerland; 4Unit of Obstetrics and Gynecology, Ospedale Villa Scassi-ASL3, Metropolitan Area of Genoa, 16149 Genoa, Italy; 5Clinical Research and Epidemiology Department, Public Health Center, Hospices Civils de Lyon, F-69003 Lyon, France; 6Breast, Gynecology and Reconstructive Surgery Unit, Institut Curie, Paris University, F-75005 Paris, France; 7Department of Obstetrics and Gynecology, Felix Guyon Hospital, University Hospital La Réunion, F-97490 Saint-Denis, France; 8University of La Réunion, F-97744 Saint-Denis, France

**Keywords:** cervical cancer, sentinel lymph node, pelvic lymphadenectomy, SENTICOL, ultrastaging

## Abstract

**Simple Summary:**

For lymph node staging in the early stages of cervical cancer, sentinel lymph node (SLN) assessment, instead of full pelvic lymph node dissection, decreased the postoperative morbidity compared to full pelvic lymph node dissection. Bilateral negative SLN predicts the absence of nodal metastases, but the risk factors for lymph node involvement beyond a positive SLN remain poorly described. Through a pooled analysis of 405 patients with early cervical cancers issued from prospective multicentric cohorts SENTICOL I and II, age and lympho-vascular invasion were retained as predictors of metastasis in non-SLN patients with SLN metastasis in early-stage cervical cancer.

**Abstract:**

Background: The goal of this study was to identify the risk factors for metastasis in the remaining non-sentinel lymph nodes (SLN) in the case of positive SLN in early-stage cervical cancer. Methods: An ancillary analysis of two prospective multicentric databases on SLN biopsy for cervical cancer (SENTICOL I and II) was performed. Patients with early-stage cervical cancer (FIGO 2018 IA to IIA1), with bilateral SLN detection and at least one positive SLN after ultrastaging, were included. Results: 405 patients were included in SENTICOL I and Il. Fifty-two patients had bilateral SLN detection and were found to have SLN metastasis. After pelvic lymphadenectomy, metastatic involvement of non-SLN was diagnosed in 7 patients (13.5%). Patients with metastatic non-SLN were older (51.9 vs. 40.8 years, *p* = 0.01), had more often lympho-vascular space invasion (LVSI) (85.7% vs. 35.6%, *p* = 0.03), and had more often parametrial involvement (42.9% vs. 6.7%, *p* = 0.003). Multivariate analysis retained age (OR = 1.16, 95% IC = [1.01–1.32], *p* = 0.03) and LVSI (OR = 25.97, 95% IC = [1.16–582.1], *p* = 0.04) as independently associated with non-SLN involvement. Conclusions: Age and LVSI seemed to be predictive of non-SLN metastasis in patients with SLN metastasis in early-stage cervical cancer. Larger cohorts are needed to confirm the results and clinical usefulness of such findings.

## 1. Introduction

Globally, cervical cancer represents the fourth most common cancer worldwide. Despite the improvement of cervical screening and the development of HPV vaccination, 600,000 cases remain diagnosed annually, resulting in 340,000 deaths [1,2]. Lymph node involvement is considered a major prognostic factor justifying its integration in the last FIGO 2018 classification [3,4,5]. Historically, the association of radical hysterectomy and pelvic-lymph-node dissection (PLND) was the gold standard of treatment for early-stage cervical cancer with negative lymph nodes on radiological staging (International Federation of Gynecology and Obstetrics (FIGO) 2018 classification stages IB1 and IIA1) [6]. However, the PLND carries a significant burden in terms of morbidity, especially lower limb lymphedema. Indeed, lymph nodes are involved in approximately 19% of patients with early-stage disease, leading to overtreatment with unnecessary PLND in most cases [7,8,9]. For the last two decades, lymph node staging for early-stage cervical cancer took a new turn with the application of the sentinel lymph node concept previously described in breast and vulvar cancer. The feasibility and safety of SLNB have been demonstrated in several studies [6,8,10,11,12]. This technique underlined that the lymphatic drainage pathway may be aberrant and that the first lymph node draining the cervix might not be removed during routine PLND [13]. Several studies demonstrated that SLN biopsy could provide the same information as PLND with less morbidity [11], better quality of life [14], and without compromising oncologic outcomes [15,16,17]. Most of the guidelines nowadays recommend the SLNB in the early stages of cervical cancer [6,18].

The SENTICOL I study showed that a bilateral negative SLN was predictive of the absence of lymph node involvement in the remaining pelvic lymph nodes [10,19]. By contrast, a positive SLN does not predict the status of downstream lymph nodes. Furthermore, according to current guidelines and the ABRAX study [20], metastatic SLN diagnosed at frozen section analysis implies that radical hysterectomy should be abandoned and an additional paraaortic LND for nodal staging purposes may be performed before the patient is referred for definitive chemo-radiation.

However, the number of metastatic nodes per patient ranges from 1 to 3, with only one node being involved in nearly half of pN1 patients which is the above-mentioned SLN [21,22,23,24,25]. This consideration raises the question of adequately identifying the patients who might benefit from the additional lymph node dissection.

The aim of this study was to identify the risk factors for metastasis in non-SLN cases of positive SLNB in patients with early-stage cervical cancer.

## 2. Materials and Methods

### 2.1. Cohort Description

Ancillary analysis of two prospective multicentric trials on Sentinel Lymph Node (SLN) biopsy for early-stage cervical cancer (SENTICOL I and SENTICOL II) was carried out. The design of both studies has already been described elsewhere [8,9]. Briefly, in SENTICOL I, a systematic pelvic lymphadenectomy was performed after the SLN biopsy, whatever the results of the FSE. In SENTICOL II, patients with bilateral negative SLN by FSE were randomized into two groups: group A, SLN biopsy alone, and group B, SLN biopsy with additional pelvic lymphadenectomy.

In both studies, all patients had early-stage cervical cancer up to the IIA1 FIGO 2018 classification (except IB3), no suspicious nodes at preoperative pelvic MRI, and underwent SLN mapping.

Patients who had bilateral SLN detection followed by full pelvic lymphadenectomy and at least one positive SLN after ultrastaging were included in the present study.

This study was approved by the Institutional Review Board of Paris Descartes (HEGP-Broussais) (ethical code: DRRC AOR 03063) and Lyon’s Civil Hospices’ Ethical Committee (ethical code: 2008-A01369-46). Patients included in both studies provided written informed consent stating the use of data for secondary analysis.

### 2.2. Data Analysis

From both databases, patient demographics, tumor characteristics, and pathologic reports were extracted and analyzed. Pathology reports were reviewed, and the FIGO stage was modified according to the 2018 FIGO classification. Tumor size was macroscopically measured in surgical specimens. The presence of LVSI was defined by the presence of tumor cells in the lumen of vessels or lymphatic channels.

Sentinel lymph nodes were detected by a combined labeling technique. The radioactive tracer (colloidal rhenium sulfide labeled with technetium (^99m^Tc)) was injected using a 25-gauge needle into the four cardinal points of the uterine cervix either on the day before surgery (120 MBq) as a long protocol or the morning of surgery (60 MBq) as a short protocol. Under general anesthesia, 2.5% Patent Blue (2 mL diluted in 2 mL of saline) was additionally injected in the same way as the radiotracer.

Frozen section analysis was performed on a sentinel lymph node biopsy, either routinely or only on suspected metastasis nodes at the surgeon’s discretion. SLN were subjected to ultrastaging. SLNs were analyzed after hematoxylin and eosin staining of 200 µm sections. Negative SLNs were then examined by immunohistochemistry with anti-cytokeratin AE1-AE3 antibodies. Positive SLN was defined as macrometastases (MAC) (tumor deposit greater than 2 mm), micrometastases (MIC) (tumor deposit greater than 0.2 and up to 2 mm), and isolated tumor cells (ITC) (tumor deposit no greater than 0.2 mm) [26]. In the same way, non-SLNs were also subjected to ultrastaging.

### 2.3. Statistics

Patients were categorized into two groups according to the presence or absence of non-SLN involvement. Qualitative variables were expressed as *n* (%) and were compared by applying the chi-square test. Quantitative data were expressed as the mean ± standard deviation (SE) or as median [range] and were compared by applying the Student’s *t*-test or a Wilcoxon test in cases of non-parametric distribution. Variables yielding *p*-values lower than 0.2 by univariate analysis were entered into a multivariate logistic regression model to determine variables independently associated with positive non-SLN. *p* values lower than 0.05 were considered significant.

All statistical analyses were run using JASP (Version 0.17.1)

## 3. Results

Overall, 405 patients were included in the SENTICOL I and II studies. Of those, 228 patients had bilateral SLN detection associated with bilateral pelvic lymphadenectomy. Finally, 52 patients had at least one positive SLN and were included for analysis. The flow chart is reported in Figure 1.

Among these 52 patients, 24 (46.2%) were found to have macrometastasis, 14 (26.9%) with micrometastases, and 14 (26.9%) with isolated tumor cells. The mean age of the population study was 42.2 years old (±10.4 years) and the mean BMI was 23.1 kg/m^2^ (±4.77 kg/m^2^). Most of the patients (67.3%) had FIGO IB1 2018 clinical FIGO stage of cervical cancer. Histology showed squamous cell carcinoma in 80.8% of patients, whereas 19.2% had adenocarcinoma. Characteristics of the population study are described in Table 1.

Overall, 177 SLNs were retrieved: 98 SLNs (55.4%) were “blue” and “hot”, 53 SLNs (30%) were “hot” only, and 26 SLNs (14.7%) were “blue” only. The mean number of SLNs found per patient was 4.00 ± 1.78. Among these 177 SLNs, 66 were metastatic: 33 had macrometastases, 19 had micrometastases, and 17 had ITCs. The mean number of positive SLNs found per patient was 1.36 ± 0.66. The topography of SLNs is reported in Table 2.

After pelvic and paraaortic lymphadenectomy, a mean of 20.6 ± 12.0 lymphatic nodes were removed per patient. Metastatic involvement of non-SLN was diagnosed in 7 of 52 patients (13.5%), whereas SLN were the only positive nodes in 45 patients (86.5%). Descriptions of the seven patients with positive non-SLN are reported in Table 3.

Patients with metastatic non-SLN were more likely to be older (51.9 vs. 40.8 years, *p* = 0.01), have more lymphovascular space invasion (LVSI) (85.7% vs. 35.6%, *p* = 0.03), and have more parametrial involvement (42.9% vs. 6.7%, *p* = 0.003). The number of positive SLN tended to be higher for patients with positive non-SLN (1.86 vs. 1.29, *p* = 0.07) without reaching the statistical significance level. Between both groups, there were no differences in terms of preoperative brachytherapy and preoperative conization (Table 4).

Multivariate analysis retained age (OR = 1.16, 95% IC = [1.01–1.32], *p* = 0.03) and LVSI (OR = 25.97, 95% IC = [1.16–582.101], *p* = 0.04) as independently associated with non-SLN involvement (Table 5).

## 4. Discussion

In the present study, for patients with SLN metastasis in the early stages of cervical cancer, age, and LVSI are independently associated with non-SLN involvement during PLND.

SLN biopsy reduces morbidity in patients with early stages of cervical cancer without jeopardizing oncologic outcomes. This assertion has been based on SENTICOL I and II trials [10,17], but also on the SENTIX trial [24] and others [27,28,29]. SLN biopsy rather than PLND carries a lower risk of lymphocele and lymphedema, shorter operative time, and fewer intraoperative complications (bladder, bowel, nerves, and vessels, amongst others) [14]. The international guidelines depict the growing role of the SLN in each stage of cervical cancer in the current process of decreasing surgical aggressiveness [30]. For instance, the updated 2023 European Society of Gynecological Oncology (ESGO) guidelines, based part on the Memorial Sloan Kettering Cancer Center (MSKCC) algorithm, give a central place to SLN in the early stages of cancer. The updated guidelines from 2023 recommend SLN alone for T1a1 with LVSI and T1a2 without LVSI. For T1b1, T1b2, and T2a1 with negative LN on radiological staging, an SLN biopsy should be performed before pelvic lymphadenectomy. This is important, as the SLND strategy is not yet validated as a standard of care for cervical cancer [31]. The American National Comprehensive Cancer Network (NCCN) guidelines recommend pelvic lymphadenectomy (with or without SLN biopsy) for 2018 FIGO stage IA2 (they however agree for SLND only on FIGO 2018 stage IA1, and for IB1 FIGO 2018 and higher to proceed with systematic pelvic lymphadenectomy +/− SLND). Differences in management between societies highlight the lack of strong evidence rallying everyone [18]. For both of them, when a tumor is found during an intraoperative frozen section, PLND and radical hysterectomy should be avoided. This recommendation underlines the importance of correctly identifying patients with LN involvement, whether SLN or non-SLN. Indeed, treatment modalities change drastically if PLN is involved or not.

The SENTICOL I study demonstrated that bilateral negative sentinel nodes accurately predicted the absence of downstream lymph node metastasis in early cervical cancers [10,12]. However, the presence of positive SLN does not predict the status of non-SLN downstream. In the present study, age and the presence of LVSI are independently associated with positive non-SLN. Clinically, these two factors can be determined preoperatively, but only under certain conditions. For instance, LVSI are rarely found on a biopsy alone (depending on the location of the cervical biopsy and whether the tumor has massive LVSI), whereas, on a conization specimen, LVSI are more easily visualized [32].

In current practice, SLN status is assessed by ultrastaging, whereas non-SLN status is assessed only through “gross” histology, leading to missed MICs and ITCs. Indeed, the current research disregards the role of MICs and ITCs, which can go unnoticed in non-SLN by standard LN histological evaluation. Ultrastaging for PLND + SLN does not seem to enhance metastasis detection, but results might be due to the low prevalence of MIC in low-risk patients and results extrapolated from a small series of patients [33,34]. If the ultrastaging increases the probability of finding smaller metastases such as MICs and ITCs, the impact of such findings is nowadays a subject of debate. A meta-analysis showed that MICs have the same clinical impacts as MACs. However, the prognostic impact of ITCs on patients with cervical cancer remains unclear [35,36]. One of the strengths of the present study is that both SLN and non-SLN were analyzed with ultrastaging. Other authors have studied ultrastaging on SLN and non-SLN, such as Cibula et al., in a cohort of 17 patients with FIGO stage IB-IIA. Two patients over 17 had MIC in the non-SLN [26]. Popa et al., with a comparable study design, found no metastasis in non-SLN when SLN was negative on final pathology in their 36 patients [37]. Outcomes safety regarding SLN sampling versus standard PLND must be of main concern when implementing a new procedure. Although it was not analyzed in the present study, a meta-analysis on disease-free survival and other oncological outcomes was recently conducted by Parpinel et al., disclosing no differences in survival and 5-year disease-free survival between SLN mapping and pelvic lymphadenectomy [38]. Regarding the safety of the technique, a recent review from Chiyoda et al. found out that sentinel lymph node navigation surgery with Technetium (with or without blue dye) or ICG was safe in SLN assessment (reducing surgical complications and improving SLN detection), but without relevant information regarding the long-term safety of SLN harvest [39]. Adequately detecting the three types of metastases (MAC-MIC-ITCs) is of importance since MICs and MACs have clinical impacts (but the prognostic impact of ITCs remains unclear), as a recent meta-analysis disclosed [35].

With pathology being considered the gold standard in the detection of metastasis, it therefore gives the exact representation of tumor load in all LN. This is relevant in view of the fact that, in current practice, a negative SLN at frozen section examination (FSE) implies that PLND is not carried out. However, an LN with MIC may be missed at the initial evaluation. If, on the other hand, the extemporaneous test is positive (=macrometastasis), one will perform the PLND. As a reminder, FSE is recommended in the guidelines but carries some intrinsic limitations. A retrospective study retained a sensitivity of 81% and an NPV of 97.9% of FSE (with ITCs being excluded from the study) [40]. The SENTIX trial found that FSE failed to detect 54% of positive lymph nodes, especially MIC (90% of cases) [24]. The same results were found in SENTICOL’s cohort, with a low sensitivity of FSE for detecting MICs and ITCs [41]. Thus, developing strategies with new technologies to enhance MIC detection is under consideration, such as mRNA amplification of tissues during intra-operative SLN biopsies. In this subject, the OSNA study found a false negative rate of 14.3% for detecting lymph node metastasis, even with micro-metastasis [42]. The rationale for such a demanding procedure is supported by data from two large retrospective studies confirming that the presence of MIC was associated with a significant negative impact on survival, which is comparable to patients with MAC [36,43]. Furthermore, missing a MIC in the SLN assessment would lead to a radical hysterectomy, which is currently not indicated based on the ABRAX study and the ESGO/ESTRO/ESP 2023 guidelines [20,31]. The debate remains open on the prognostic importance of ITC. As previously stated, the impact of positive SLNB has been demonstrated, but the impact of the positive non-sentinel lymph node (non-SLN) has not, as their specific place in the lymphatic involvement had previously been incorporated into the PLN involvement as a “whole”.

The status of non-SLN in cases of pelvic-positive SLN has already been investigated in endometrial cancer. In fact, in endometrial cancers, the size of the SLN metastasis is predictive of the involvement of non-SLNs [44,45], which is not the case in cervical cancer. Of note, a recent retrospective study including 967 patients with FIGO T1a1L1-T2b cancers found out that the size itself of the metastasis did not impact disease-free survival (DFS) (but having a (+)SLN was of course negative on DFS) [46]. On this subject, our results are consistent with those of Diniz et al. and the literature [47]. These results raised the question of the specificity of lymphatic spread in cervical cancer, and we support the idea that cervical cancer spreads differently from endometrial cancer in the lymphatic system. The particularity of cervical cancer is lymphophilia and parametrial invasion (hence the interest in parametrectomy of the radical hysterectomy). The parametrial spread of cervical cancer was supported by several previous studies [22,48,49,50]. In our study, this result was positive in univariate analysis. However, it was not retained in the multivariate analysis. This is most probably due to a lack of cases and, therefore, of power. Our univariate results are corroborated by Diniz’s study [47]. They retained parametrial invasion as the only risk factor for positive non-SLN (*p* = 0.045).

Cervical cancer is a pathology from the midline of the body; thus, the spread to the LN can be bilateral or unilateral. Furthermore, in addition to anatomical consideration, bilateral surgical detection of LN is lower than unilateral (i.e., 72% vs. 95% detection rate [51]). As previously stated in the SENTICOL II cohort, the LN detection rate was 76.5% (in line with the literature), and for bilateral SLN detection, the negative predictive value (NPV) was 100% with no false negatives (FNs) [12,13].

Regarding LVSI+ as a risk factor for subsequent non-SLN(+) in early cervical cancers, the trend is in line with the known risk in the literature, as many studies have shown [9,52,53]. LVSI is associated with a high risk of recurrences and decreased 5-year disease-free survival in the early stages [54]. Stromal invasion is also a parameter that should drive the extension of the surgery. It was not evaluated in the present study due to missing data.

Patients aged 50 years and older exhibited a higher risk of recurrences, aligning with previous research [55,56,57]. This disparity was attributed to older patients presenting with more advanced disease stages, often due to reduced gynecologic follow-up and delayed disease detection. In addition, it is plausible that age and/or co-morbidities may have influenced older patients’ receipt of suboptimal surgical and/or adjuvant treatment. Moreover, a trial conducted by Quinn et al. showed that older women with cervical cancer have poorer overall survival regardless of the stage of the disease at presentation or the histological type, assuming that advancing age is to be considered an independent negative prognostic factor. The correlation between increasing age as a risk factor for non-SLN(+) metastasis is still unclear [58]. Anatomical study to assess how the lymphatic drainage system varies and is more “porous” over time has been studied. Balaya et al. showed that age over 70 was an independent risk factor for failure of SLN mapping [15]. And so did Seong et al., with an age limit of 50 years [59]. The aging process of lymphatic vessels might as well be part of the process of modifying drainage routes [60].

The present study has some limitations. The first is the SENTICOL I and II design, as it was not primarily designed for the primary outcome of this study. On the other hand, the prospective multicentric design gives supplementary strength to the data. Moreover, whereas other pelvic LNs were handled according to standard procedures (one section), SLNs were evaluated using a pathological ultrastaging approach. In order to compare SLN and non-SLN status, one can only acknowledge the fact that there is a slight methodological bias. As previously stated, some studies have assessed SLNs and all other PLNs by ultrastaging for both groups, with mixed results regarding false negative rates, but the cohort was relatively small [33,34,37]. Selecting only patients with bilateral SLN detection naturally reduces the number of patients that can be included, as previously stated. The strengths of this study are the prospective design, in addition to the relatively large number of patients included in multiple surgical centers and, more importantly, the ultrastaging performed in all the LN (SLN and non-SLN). Our findings stress the importance of comprehension of risk factors for invasion by non-SLN, as they might influence the treatment strategy.

Although future prospective studies are, for instance, aiming to compare survival and health-related quality of life after SLN biopsy or SLN biopsy and pelvic lymphadenectomy in early cervical cancer, such as the SENTICOL III study [61] or the oncological outcomes of SLN biopsy with pelvic lymphadenectomy in patients with and without SLN metastasis (PHENIX/CSEM study) [62], there is little place in the literature and ongoing research to evaluate the role of the positive non-SLN in clinical outcomes. Therefore, future research should focus on prospective assessment of the non-SLN, preferably with ultrastaging to avoid missing the MICs and ITCs, and to evaluate the clinical outcomes of such patients, maybe in order to tailor treatments.

## 5. Conclusions

Older age and LVSI seemed to be predictive of non-sentinel lymph node metastasis in patients with sentinel lymph node metastasis in early-stage cervical cancer. Further studies with a larger cohort are required to confirm these results and confirm the clinical usefulness of such findings.

## Figures and Tables

**Figure 1 cancers-15-04737-f001:**
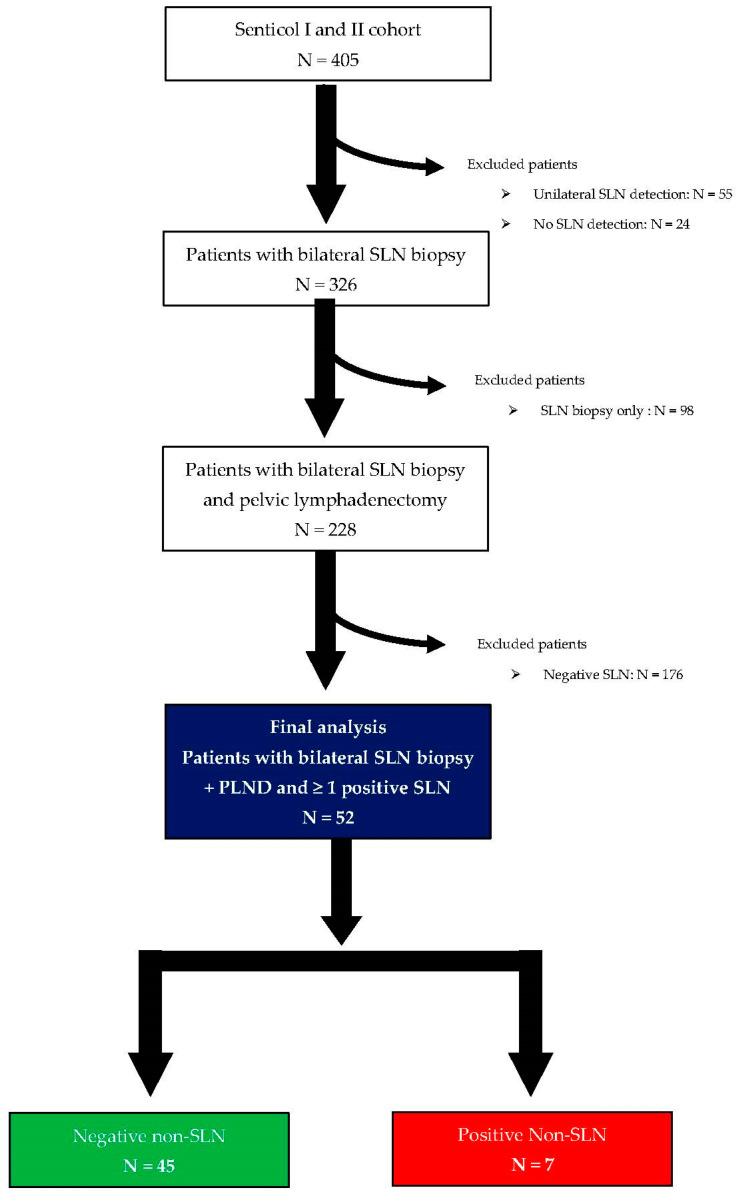
Flowchart of patient’s selection process within SENTICOL I and II.

**Table 1 cancers-15-04737-t001:** Population characteristics.

Predictive Variable	Total Population N = 52	
	* n * Mean ± SD	[%] [Range]
**Age [years]**		
**Mean**	42.2 ± 10.4	[25–77]
**BMI [kg/m^2^]**		
**Mean**	23.1 ± 4.77	[17.0–38.8]
**Parity status**		
**0**	14	26.9
**≥1**	38	73.1
**Histology**		
**Squamous cell carcinoma**	42	80.8
**Adenocarcinoma**	10	19.2
**Preoperative brachytherapy**		
**Yes**	10	23.3
**No**	33	76.7
**Not specified**	9	
**Preoperative conization**		
**Yes**	22	42.3
**No**	30	57.7
**Clinical 2018 FIGO stage**		
**IA**	3	5.8
**IB1**	35	67.3
**IB2**	13	25.0
**IIA**	1	1.9
**SLN status**		
**Macrometastases**	24	46.2
**Micrometastases**	14	26.9
**ITCs**	14	26.9
**Bilateral SLN involvement**		
**Yes**	11	21.2
**No**	41	78.8
**Number of positive SLN**	1.36 ± 0.66	[1–4]
**Tumor size**		
**Mean (mm)**	22.8 ± 13.4	[4–70]
**LVSI**		
**Yes**	22	42.3
**No**	30	57.7
**Parametrial invasion**		
**Yes**	6	11.5
**No**	46	88.5
**Vaginal invasion**		
**Yes**	5	9.6
**No**	47	90.4

SLN: sentinel lymph node, BMI: body mass index, FIGO: International Federation of Gynecology and Obstetrics, ITCs: isolated tumor cells, LVSI: lympho-vascular invasion.

**Table 2 cancers-15-04737-t002:** SLN status and topography.

Variable	Overall SLN		Overall Positive SLN		MAC		MIC		ITCs		*p*
	* n *	[%]	* n *	[%]	* n *	[%]	*n*	[%]	*n*	[%]	
**Topography**											
**Interiliac/External iliac area**	141	79.7	58	87.9	25	83.3	17	89.5	16	94.1	*0.74*
**Common iliac area**	18	10.2	3	4.6	-	-	2	10.5	1	5.9	
**Parametrial area**	8	4.5	3	4.6	3	10	-	-	-	-	
**Promontory area**	6	3.4	1	1.5	1	3.3	-	-	-	-	
**Paraaortic area**	4	2.3	1	1.5	1	3.3	-	-	-	-	
**Total**	177	100	66	100	30	100	19	100	17	100	

SLN: sentinel lymph node, MAC: macrometastasis, MIC: micrometastasis, ITCs: isolated tumor cells.

**Table 3 cancers-15-04737-t003:** Clinicopathologic characteristics of patients with non-SLNs involvement.

Patient	Age	Histologic Type	Clinical 2018 FIGO Stage	Presence of LVSI	Parametrial Involvement	Number of Metastatic SLNs/Total SLNs	Location of Involved SLNs	Type of SLNs Involvement	Number of Metastatic Non-SLNs/Total Non-SLNs	Location of Involved Non-SLNs	Type of Non-SLNs Involvement
**1**	77	Squamous cell carcinoma	IB2	Yes	Yes	1/2	Left interiliac	ITCs	4/25	Right interiliac (3) Right common iliac (1)	MIC and ITCs
**2**	49	Squamous cell carcinoma	IB2	Yes	Yes	3/5	Right interiliac Left Parametrium Left interiliac	ITCs MAC MAC	1/19	Right interiliac	MAC
**3**	45	Squamous cell carcinoma	IB2	Yes	Yes	2/2	Right interiliac Left interiliac	ITCs MAC	1/7	Left interiliac	MAC
**4**	48	Squamous cell carcinoma	IB1	Yes	No	2/9	Right interiliac Left interiliac	MIC MIC	1/18	Right interiliac	MAC
**5**	42	Squamous cell carcinoma	IB1	Yes	No	1/2	Right interiliac	MAC	2/39	-	-
**6**	49	Squamous cell carcinoma	IB1	No	No	1/2	Left interiliac	MIC	1/26	Left paraaortic	MAC
**7**	53	Squamous cell carcinoma	IB2	Yes	No	1/3	Right interiliac	MAC	6/26	Left interiliac (1) Right interiliac (5)	MAC MAC

SLN: sentinel lymph node, BMI: body mass index, FIGO: International Federation of Gynecology and Obstetrics, MAC: macrometastasis, MIC: micrometastasis, ITCs: isolated tumor cells, LVSI: lympho-vascular invasion.

**Table 4 cancers-15-04737-t004:** Univariate analysis of predictive factors of non-SLN involvement.

Predictive Variable	Patient with Negative Non-SLN N = 45		Patient with Positive Non-SLN N = 7		* p *
	* n * Mean ± SD	[%] [Range]	* n * Mean ± SD	[%] [Range]	
**Age [years]**					
**Mean**	40.8 ± 9.5	[25–64]	51.9 ± 11.6	[42–77]	0.01
**BMI [kg/m^2^]**					
**Mean**	22.6 ± 4.2	[17.0–33.7]	26.5 ± 7.0	[19.3–38.8]	0.11
**Parity status**					
**0**	13	28.9	1	14.3	0.66
**≥1**	32	71.1	6	85.7	
**Histology**					
**Squamous cell carcinoma**	35	77.8	7	100.0	0.32
**Adenocarcinoma**	10	22.2	0	0.0	
**Preoperative brachytherapy**					
**Yes**	8	21.1	2	40.0	0.57
**No**	30	78.9	3	60.0	
**Not specified**	7		2		
**Preoperative conization**					
**Yes**	20	44.4	2	28.6	0.68
**No**	25	55.6	5	9.6	
**Clinical 2018 FIGO stage**					
**IA**	3	6.7	0	0.0	0.23
**IB1**	32	71.1	3	42.9	
**IB2**	9	20.0	4	57.1	
**IIA**	1	2.2	0	0.0	
**SLN status**					
**Macrometastases**	20	44.4	4	57.1	0.87
**Micrometastases**	12	26.7	2	28.6	
**ITCs**	13	28.9	1	14.3	
**Bilateral SLN involvement**					
**Yes**	8	17.8	3	42.9	0.15
**No**	37	82.2	4	57.1	
**Number of positive SLN**	1.29 ± 0.55	[1–3]	1.86 ± 1.07	[1–4]	0.07
**Tumor size**					
**Mean (mm)**	21.9 ± 13.0	[4–70]	28.1 ± 15.4	[7–51]	0.20
**LVSI**					
**Yes**	16	35.6	6	85.7	0.03
**No**	29	64.4	1	14.3	
**Parametrial invasion**					
**Yes**	3	6.7	3	42.9	0.03
**No**	42	93.3	4	57.1	
**Vaginal invasion**					
**Yes**	3	6.7	2	28.6	0.13
**No**	42	93.3	5	9.6	

SLN: sentinel lymph node, BMI: body mass index, FIGO: International Federation of Gynecology and Obstetrics, ITCs: isolated tumor cells, LVSI: lympho-vascular invasion. *p*-value in red is described as significant (*p* < 0.05).

**Table 5 cancers-15-04737-t005:** Multivariate analysis of factors associated with successful bilateral SLN mapping per patient.

Variable	ORa	IC 95%	* p *
**Age [years]**			
	**1.16**	**1.01–1.32**	** 0.03 **
**BMI (kg/m^2^)**			
	1.25	0.94–1.66	0.12
**Bilateral SLN involvement**			
**No**	1		
**Yes**	2.65	0.01–523.29	0.72
**Number of positive SLN**			
	0.60	0.01–38.01	0.81
**LVSI**			
**No**	**1**		
**Yes**	**25.97**	**1.16–582.1**	** 0.04 **
**Parametrial invasion**			
**No**	1		
**Yes**	1.20	0.01–121.64	0.94
**Vaginal invasion**			
**No**	1		
**Yes**	0.65	0.01–117.25	0.87

SLN: sentinel lymph node, BMI: body mass index, LVSI: lympho-vascular invasion. OR: odds ratio, IC: interval of confidence, *p*-value in red is described as significant (*p* < 0.05).

## Data Availability

The data presented in this study are available on reasonable request from the corresponding author. The data are not publicly available due to data privacy.

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
