# Peer review of "Predictors of Non-Sentinel Lymph Node Metastasis in Patients with Positive Sentinel Lymph Node in Early-Stage Cervical Cancer: A SENTICOL GROUP Study"

_cancers, 2023, doi:10.3390/cancers15194737_

Round 1

Reviewer 1 Report

General

The authors present an excellent study about a scientifically and clinical important question. We know well the diagnostic value of a negative sentinel lymph node in the treatment of cervical carcinoma, but more less about the significance of a positive sentinel biopsy.

The authors use a multicentric prospective data base to give new information for this critical question.

They present their data carefully and transparent and discuss their results critically.

Therefore I have only few minor suggestions.

Minor points

Line 38: “more” seems to be a little unprecise: “more often”

Line 29:     -“-

Line 111: This statement refers to sentinel lymph node biopsy?

Line 163: Fig.1. The number N=32610 is obviously wrong.

Lin3 382: “older age” would be more clear.

Author Response

Dear reviewer,

Thank you very much for your thorough review. Please find hereafter the comments to your revision.

Best regards,

Basile PACHE

  • Line 38: “more” seems to be a little unprecise: “more often”

                                                We modified accordingly

  • Line 29: -“-

                                                We modified accordingly

  • Line 111: This statement refers to sentinel lymph node biopsy?

Yes indeed. We modified the sentence accordingly. “Frozen section analysis was performed on sentinel lymph node biopsy either routinely or only on suspected metastasis nodes at the surgeon discretion.”

  • Line 163: Fig.1. The number N=32610 is obviously wrong.

Sorry for the typo, we corrected to “326”.

  • Line 382: “older age” would be more clear

                                         We modified accordingly

Reviewer 2 Report

I read with great interest this Manuscript, which falls within the aim of the Journal.Honestly, the topic is interesting enough to attract the readers’ attention considering the growing interest in the role of SLN in Early Cervical Cancer. The methodology is accurate, the data analysis supports the conclusions, strengths and limitations of the study are clearly defined. I have no particular suggestions for this work, except the possibility for authors to specify any correlations ( if any )  between the localization of the  positive non-sentinel node   and  site and  type  (MAC,MIC , ITCS) of metastasis in the sentinel node analyzed in order to define if  LVSI as risk factor in non-sentinel positive node could justify any changes in lymphatic drainage patterns , also  considering   its useful evaluation to identify risk patients for shorter disease-free survival and lymphatic and distant recurrences ( PLS see 10.1016/j.ygyno.2021.06.002)

In addition, considering the innovative character of the subject and the debated role of exclusive SLN mapping in ECC, I suggest the authors to implement discussion  by comparing the issues addressed by  new  works already  exist in literature(PLS see 10.1007/s10147-022-02178-w /10.1007/s10147-022-02178-w)

Some text editing refuses.

Author Response

Dear reviewer,

Thank you very much for your thorough review. Please find hereafter the comments to your revision.

Best regards,

Basile PACHE

  • I read with great interest this Manuscript, which falls within the aim of the Journal. Honestly, the topic is interesting enough to attract the readers’ attention considering the growing interest in the role of SLN in Early Cervical Cancer. The methodology is accurate, the data analysis supports the conclusions, strengths and limitations of the study are clearly defined. I have no particular suggestions for this work, except the possibility for authors to specify any correlations ( if any ) between the localization of the  positive non-sentinel node   and  site and  type  (MAC,MIC , ITCS) of metastasis in the sentinel node analyzed in order to define if  LVSI as risk factor in non-sentinel positive node could justify any changes in lymphatic drainage patterns , also  considering   its useful evaluation to identify risk patients for shorter disease-free survival and lymphatic and distant recurrences ( https://pubmed.ncbi.nlm.nih.gov/34116834/)

Dear reviewer, thank you for your kind comments, but especially thank you for bringing up the subject of the localization pattern of subtype of SLN metastasis. We indeed reviewed in Table 2 the SLN status and topography, and in Table 4 we depicted the 7 cases of patients with positive non-SLN. This number is unfortunately too small to drive a specific correlation.

  • In addition, considering the innovative character of the subject and the debated role of exclusive SLN mapping in ECC, I suggest the authors to implement discussion by comparing the issues addressed by new works already existing in literature (https://pubmed.ncbi.nlm.nih.gov/35612720/)

The systematic review and meta-analysis of Tatsuyuki Chiyoda reviewed sentinel node navigation surgery in cervical cancer. Although we did not analyze the clinical outcomes of patients with/without nonSLN, we added a paragraph in the discussion as follow:

“Outcomes safety regarding SLN sampling versus standard PLND must be of main concern when implementing a new procedure. Although it was not analyzed in the present study, a metanalysis on disease-free survival and other oncological outcomes was recently conducted by Parpinel et al, disclosing no differences in survival and 5-year disease-free between SLN mapping and pelvic lymphadenectomy. Regarding the safety of the technique, a recent review from Chiyoda et al. found out that sentinel lymph node navigation surgery with Technetium (with or without blue dye) or ICG were safe in SLN assessment (reducing surgical complications and improving SLN detection) but without relevant information regarding the long-term safety of SLN harvest. Adequately detecting the three types of metastases (MAC-MIC-ITCs) is of importance, since MICs and MACs have clinical impacts (but prognostic impact of ITCs remains unclear), as a recent metanalysis disclosed.”